# Class A scavenger receptor MARCO negatively regulates Ace expression and aldosterone production

**Conan JO O'Brien[1], Giorgio Ratti[1], Hellen Veida-Silva[1,2], Emma Haberman[1], Charles Sweeney[1], Siamon Gordon[3,4], Ana I Domingos[1]***

[1]Department of Physiology, Anatomy, & Genetics, University of Oxford, Oxford, United Kingdom; [2]Internal Medicine Department, University of Sao Paulo, São Paulo, Brazil; [3]College of Medicine, Chang Gung University, Taoyuan, Taiwan; [4]Sir William Dunn School of Pathology, University of Oxford, Oxford, United Kingdom

## eLife Assessment

O'Brien and co-authors provide **important** data demonstrating that tissue-resident macrophages can exert physiological functions and influence endocrine systems. Their model in which AMs negatively regulate aldosterone production via effects exerted in the lung is **solid**. The work will be of broad interest to cell biologists and immunologists.

***For correspondence:**
ana.domingos@dpag.ox.ac.uk

**Abstract** Aldosterone is a potent cholesterol-derived steroid hormone that plays a major role in controlling blood pressure via regulation of blood volume. The release of aldosterone is typically controlled by the renin–angiotensin–aldosterone system, situated in the adrenal glands, kidneys, and lungs. Here, we reveal that the class A scavenger receptor MARCO, expressed on alveolar macrophages, negatively regulates aldosterone production and suppresses angiotensin-converting enzyme (Ace) expression in the lungs of male mice. Collectively, our findings suggest alveolar macrophages as additional players in the renin–angiotensin–aldosterone system and introduce a novel example of interplay between the immune and endocrine systems.

## Introduction

The scavenger receptor MARCO (macrophage receptor with collagenous structure), a class A scavenger receptor, is primarily expressed on macrophages including tumour-associated macrophages and lung macrophages (*Shi et al., 2021*; *van der Laan et al., 1997*). MARCO, upregulated in response to pathogenic challenge (*van der Laan et al., 1997*), has been implicated in defence against pathogens in the lung (*Arredouani et al., 2004*), phagocytosis and clearance of tumour cells (*Xing et al., 2021*), internalisation of exosomes (*Kanno et al., 2020*), and has been touted as a potential target in anti-cancer immunotherapy (*Eisinger et al., 2020*). The class A scavenger receptors have a broad ligand specificity and similar domain structures, comprising a cytoplasmic tail, transmembrane region, spacer region, α helical coiled domain, collagenous domain, and a C-terminal cysteine-rich domain (*Plüddemann et al., 2007*). Known endogenous ligands for the receptor include, like other class A receptors, modified forms of low-density lipoprotein (*Plüddemann et al., 2007*) indicating that this receptor could be involved in the modulation of cholesterol availability. Scavenger receptors have previously been implicated in corticosteroid output from the adrenal gland. Specifically, Scavenger Receptor BI (SR-BI) was shown to regulate glucocorticoid responses via binding of serum cholesterol, the necessary substrate for adrenal corticosteroids (*Rigotti et al., 1997*; *Temel et al., 1997*; *Hoekstra et al.,*

**Figure 1.** Marco-deficient mice have elevated aldosterone and reduced serum cholesterol. (**A**) Schematic depicting the murine adrenal corticosteroid biosynthetic pathway. Plasma aldosterone and corticosterone concentrations from wild-type (WT) and Marco-deficient male (**B, C**) and female (**D, E**) mice as measured by ELISA. (**F, G**) Weights of both adrenal glands from WT or Marco-deficient male and female mice. Plasma total cholesterol levels (**H**) and relative intra-adrenal cholesterol levels, normalised to adrenal weight (**I**), in WT and Marco-deficient male mice. Data in B–H were analysed by two-tailed unpaired Student's *t*-test and are shown as average ± SEM. *p < 0.05, **p < 0.01, ****p < 0.0001, ns = not significant.

*2009*; *Ito et al., 2020*). Aldosterone, the other major adrenal cortex-derived corticosteroid, is known to be regulated by the renin–angiotensin–aldosterone system, which is situated in the adrenals, lung, and kidney (*Santos et al., 2019*). While a role for scavenger receptors in regulating glucocorticoid responses has been demonstrated, the role of macrophage-expressed scavenger receptors in the regulation of adrenal corticosteroid output at steady state has not yet been explored.

## Results

It is known that statins, cholesterol-lowering drugs have been shown to reduce aldosterone levels in humans (*Baudrand et al., 2015*; *Hornik et al., 2020*). Moreover, in vitro studies have demonstrated that cholesterol supplementation boosts the production of aldosterone from cultured cells (*Simpson et al., 1989*; *Cherradi et al., 2001*; *Kopprasch et al., 2009*). Given that the adrenal-derived mouse corticosteroids, most notably corticosterone and aldosterone, derive from cholesterol as the common precursor (*Figure 1A*), we hypothesised that cholesterol binding scavenger receptors could modulate adrenal corticosteroid output by regulating the availability of cholesterol that could feed into the steroid hormone biosynthetic pathway. To test this hypothesis, we measured the concentrations of aldosterone and corticosterone in the plasma of *Marco*$^{-/-}$ and wild-type (WT) mice. We found that male *Marco*$^{-/-}$ mice had significantly elevated levels of plasma aldosterone relative to WT mice (*Figure 1B*). In contrast, plasma corticosterone levels were not significantly altered in male mice lacking Marco (*Figure 1C*). Marco-deficient female mice did not have altered levels of aldosterone relative to WT, but plasma corticosterone was increased relative to WT counterparts (*Figure 1D, E*). We observed that the adrenal glands from Marco-deficient male mice were significantly lighter than WT controls in male but not female mice (*Figure 1F, G*). To establish whether cholesterol could explain the elevated plasma aldosterone we observe in Marco-deficient male mice, we measured the levels of total serum cholesterol and intra-adrenal cholesterol in males. We found that *Marco*$^{-/-}$ mice had reduced serum cholesterol relative to WT controls (*Figure 1H*), while the normalised levels of intra-adrenal cholesterol were similar between both mouse strains (*Figure 1I*). Taken collectively, these findings suggest that,

while Marco-deficient male mice have elevated plasma aldosterone concentrations, this is not dependent on systemic or intra-adrenal cholesterol availability. For the purposes of this paper, we focused on the phenotype evident in male mice, namely the increase in plasma aldosterone concentrations.

We next hypothesised that adrenal gland-derived *Marco* could play a role in modulating aldosterone output. Analysis of publicly available single-cell sequencing data from murine adrenal glands shows that adrenals do contain a substantial population of macrophages expressing *Ptprc* (CD45), *Adgre1* (F4/80), and *Cd68* (CD68). However, we did not detect Marco expression in this cluster of cells, nor any other cluster identified in our analyses (*Figure 2A*). This finding was corroborated by immunostaining of male murine adrenal glands, which showed CD68+ macrophages in the adrenal zona fasciculata and zona glomerulosa that did not stain positively for MARCO (*Figure 2B*). Given that the lung is another site in the RAAS axis, we postulated that Marco-expressing cells in the lung could be involved in mediating the aldosterone phenotype we observed in *Figure 1B*. Indeed, single-cell RNA-seq analysis of the murine lung shows that *Ptprc* (CD45), *Adgre1* (F4/80), and *Cd68* (CD68) expressing cells (alveolar macrophages) also express *Marco* (*Figure 2C*), a finding corroborated by immunostaining in the lung (*Figure 2C*). To further validate our single-cell sequencing and immunofluorescence data, we carried out qPCR for *Marco* in the lungs and adrenal glands from male WT and *Marco*$^{-/-}$ mice, which further demonstrated that the lung is the primary site of Marco expression in the RAAS (*Figure 2E*).

Aldosterone is a potent blood pressure-regulating hormone, the dysregulation of which can cause severe hypertension and increased cardiovascular risk. It therefore follows that its production is tightly regulated. Aldosterone biosynthesis is fundamentally regulated intra-adrenally by cytochrome P450 family members in the corticosteroid biosynthetic pathway (*Figure 1A*). We therefore tested whether altered expression of enzymes in this pathway could explain the hyperaldosteronism observed in Marco-deficient mice. *Marco*$^{-/-}$ male and female mice showed similar expression of aldosterone biosynthetic enzymes (*Star*, *Cyp11a1*, *Hsd3b1*, *Cyp11b1*, and *Cyp11b2*) as WT mice (*Figure 3A, B*). While *Cyp11b2* (aldosterone synthase) is only expressed in the adrenal zona glomerulosa, other biosynthetic enzymes essential for aldosterone production are expressed in the zona fasciculata.

CYP11B1 catalyses the conversion of 11-deoxycorticosterone to corticosterone. Corticosterone can be catalysed to aldosterone by CYP11B2. In this sense, the route via CYP11B1 is a bona fide route for the generation of aldosterone, as evidenced by the fact that *Cyp11b1* deletion in mice results in a significant reduction in aldosterone production (*Mullins et al., 2009*). This route is one that is suppressible via the suppressive action of dexamethasone on ACTH and *Cyp11b1* expression. Moreover, ACTH is a known stimulator of aldosterone (*Seely et al., 1989*; *Daidoh et al., 1995*). Dexamethasone-mediated suppression of the zona fasciculata (*Figure 3C*; *Finco et al., 2018*) was used to test whether the elevated aldosterone phenotype was zona fasciculata-dependent. Marco-deficient mice fed dexamethasone-supplemented drinking water had plasma aldosterone concentrations comparable to vehicle-treated mice (*Figure 3D*), indicating zona fasciculata activity does not contribute to elevated aldosterone levels in *Marco*$^{-/-}$ mice. Taken collectively, these findings indicate that elevated aldosterone observed in Marco-deficient mice arises extra-adrenally and can therefore be considered a form of secondary hyperaldosteronism. Kidney-derived renin is the initiating hormone in the enzymatic cascade that generates Angiotensin II, a potent stimulator of adrenal aldosterone production. We therefore compared plasma renin activity between male WT and *Marco*$^{-/-}$ mice, finding no significant differences between the two strains (*Figure 3E*).

Next, we investigated whether lung-derived angiotensin-converting enzyme (*Ace*) could explain the elevated aldosterone levels observed in *Marco*$^{-/-}$ mice. ACE in the lung catalyses the conversion of Angiotensin I to the aldosterone-stimulating peptide Angiotensin II. We carried out a qPCR test for *Ace* in the lungs of WT and *Marco*$^{-/-}$ mice in both sexes, finding that Marco-deficient male animals had elevated levels of lung *Ace* relative to WT controls in male mice only (*Figure 4A*). Immunofluorescent staining of WT and *Marco*$^{-/-}$ lungs revealed a substantially higher level of ACE protein in Marco-deficient male mice, while myeloid presence, as measured by CD68 staining, remained unchanged (*Figure 4B*). While low levels of ACE expression could be detected in CD68+ cells (data not shown), the vast majority of ACE was outside of monocytes and macrophages. We used image analysis software to quantify these changes, finding that ACE median fluorescence intensity (MFI) was significantly increased in male Marco-deficient lungs, while CD68+ myeloid cells were present in WT and knock-out animals at similar levels (*Figure 4C, D*). Myeloid cell numbers and lung ACE expression were similar in

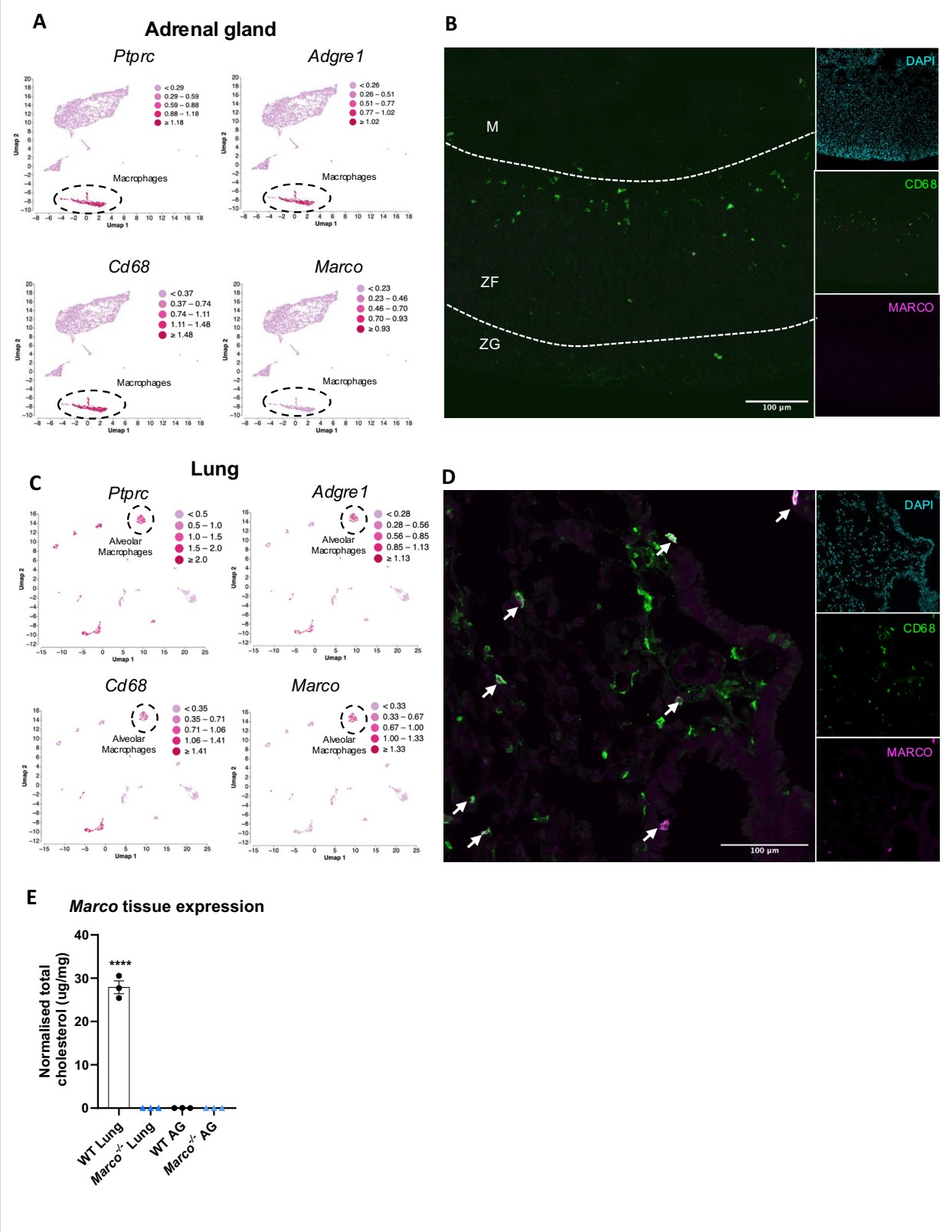

**Figure 2.** Marco is expressed in the lung and not adrenal glands. (**A**) Single-cell RNA-seq data plots from PMID 33571131 representing mRNA expression of Ptprc (CD45), Adgre1 (F4/80), Cd68, and Marco, in male murine adrenal glands. (**B**) Representative image showing a central cryosection of the male murine adrenal gland from a C57bl/6 mouse stained against CD68 (green) Marco (magenta), and DAPI (cyan). (**C**) Single-cell RNA-seq data plots from PMID 30283141 representing mRNA expression of Ptprc (CD45), Adgre1 (F4/80), Cd68, and Marco, in the male murine lung.

*Figure 2 continued on next page*

*Figure 2 continued*

(**D**) Representative image showing a cryosection of the male murine lung from a hCD68-GFP reporter mouse stained against GFP (green) Marco (magenta), and DAPI (cyan). (**E**) qPCR data showing the relative gene expression data for *Marco* in the indicated tissues. M = medulla, ZF = zona fasciculata, ZG = zona glomerulosa. Data in E were analysed by two-tailed unpaired Student's *t*-test and are shown as average ± SEM. ****p < 0.0001, ns = not significant.

WT and Marco-deficient female lungs (*Figure 4E, F*). We also measured the levels of plasma potassium and sodium levels in male and female WT and Marco-deficient animals, but observed no differences between the genotypes (*Figure 4G–J*). Since aldosterone is a known regulator of blood pressure via regulation of blood fluid balance, we also measured blood pressure using the tail-cuff method. We observed that Marco-deficient male mice had marginally reduced diastolic blood pressures, but systolic and mean blood pressure were no different between the two strains (*Figure 4K–M*).

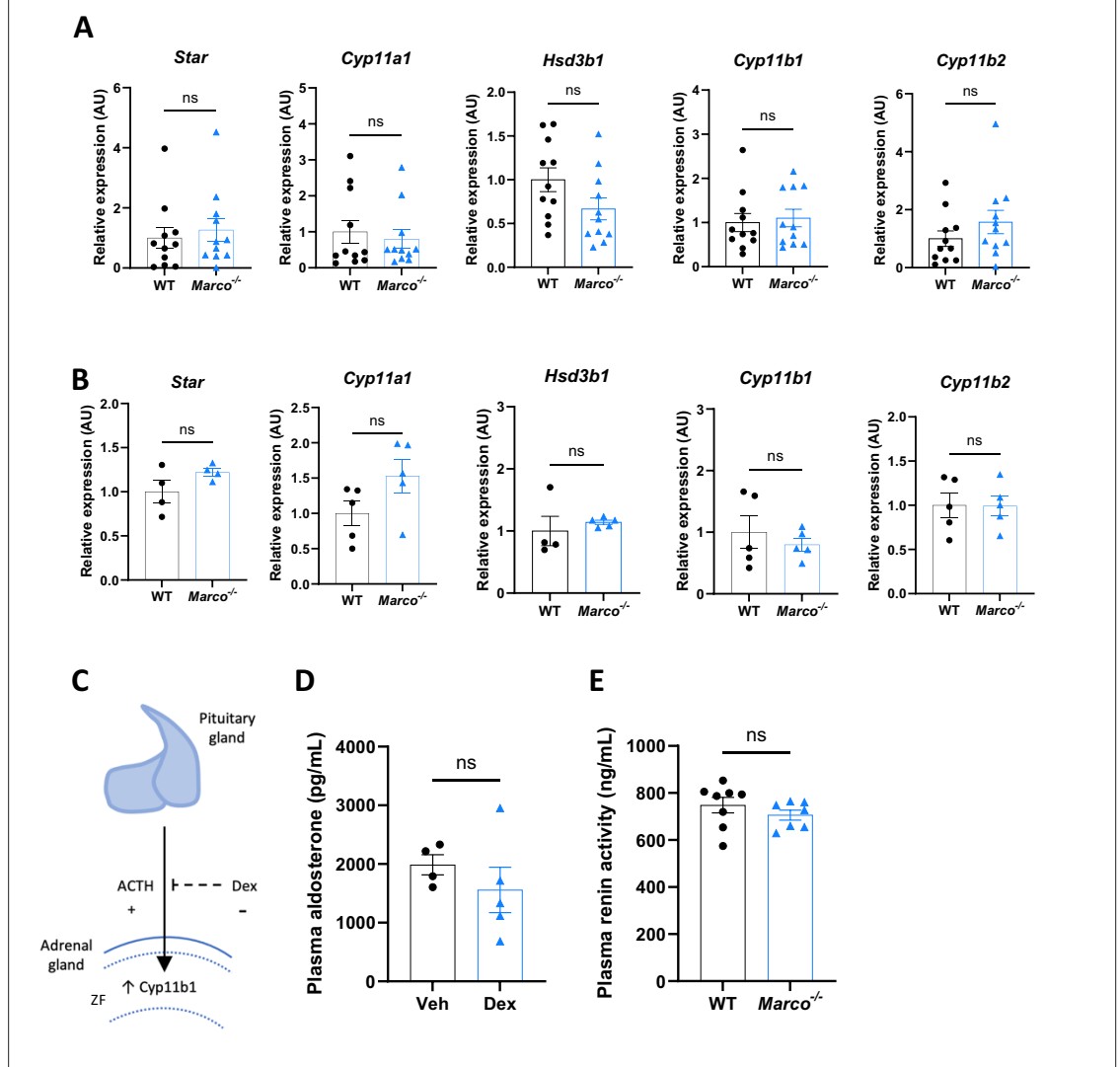

**Figure 3.** Marco-mediated elevation of aldosterone is not explained by upregulation of adrenal biosynthetic enzymes, the zona fasciculata, or renin. qPCR data reporting the expression of adrenal corticosteroid biosynthetic enzymes between wild-type (WT) and Marco-deficient male mice in male (**A**) and female (**B**) mice. (**C**) A schematic illustrating that pituitary-derived adrenocorticotropic hormone (ACTH) stimulates the upregulation of Cyp11b1 from the zona fasciculata (ZF) and the dexamethasone-mediated suppression of this effect. (**D**) A bar graph showing the plasma aldosterone concentrations of Marco-deficient mice treated with vehicle or dexamethasone-supplemented drinking water for 14 days. (**E**) A bar graph showing the plasma renin activity of WT and Marco-deficient mice at steady state. All data were analysed by two-tailed unpaired Student's *t*-test and are shown as average ± SEM. ns p > 0.05. ns = not significant.

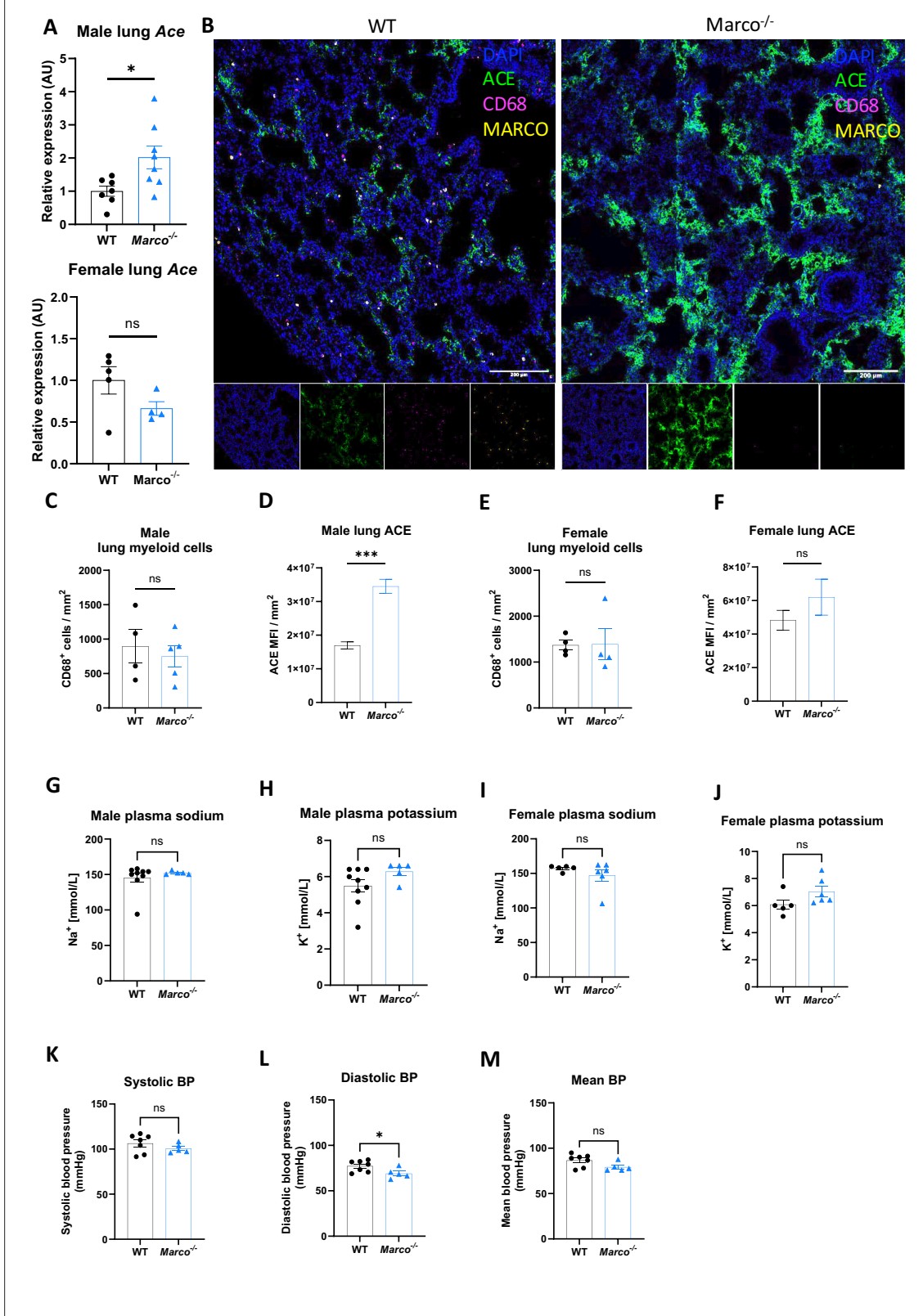

**Figure 4.** Marco-deficient mice have enhanced expression of Ace in the lung. (**A**) qPCR data reporting the expression of angiotensin-converting enzyme (*Ace*) in the lungs of wild-type (WT) and Marco-deficient male and female mice. (**B**) Representative images showing a cryosection of WT and Marco-deficient male murine lungs stained against ACE (green) CD68 (magenta), MARCO (yellow) and DAPI (blue). Quantitation of DAPI-normalised ACE median fluorescence intensity (MFI) (**C**) and CD68⁺ myeloid cell presence in the lungs of WT and Marco-deficient male and female mice (**C–F**). Plasma

*Figure 4 continued on next page*

*Figure 4 continued*

sodium and potassium levels in male and female mice (**G–J**). Systolic, diastolic, and mean blood pressure measurements in male mice of the indicated genotypes (**K–M**). Data in (**A**), (**C–M**), were analysed by two-tailed unpaired Student's *t*-test and are shown as average ± SEM. *p < 0.05, ***p < 0.001, ns = not significant.

We then turned back to analysis of single-cell RNA-seq data to identify the cells in the lung that could be mediating this effect. In the lung, unsupervised cluster analysis revealed a total of 12 cell clusters with distinct gene expression signatures (*Figure 5A*). We determined the identity of each cell cluster based on the expression of established cell type-specific marker genes, aided by the marker genes identified and outlined in *Figure 5B*. We next used dot plots to visualise expression of *Marco* and *Ace* across the different cell clusters. We observed notable *Marco* expression only in alveolar macrophages amongst the different cell clusters (*Figure 5C*). *Ace* was shown to be primarily expressed by lung endothelial clusters 1 and 2 (*Figure 5C*), in agreement with what is known about lung *Ace* expression. To test whether alveolar macrophages are capable of suppressing endothelial cell *Ace* expression, we co-cultured the MPI alveolar macrophage cell line (*Fejer et al., 2013*), with or without deletion of Marco, with HUVECs endothelial cells for 24 hr, and measured *Ace* expression via qPCR. We observed an increase in *Ace* expression in the Marco-deficient MPI co-cultures but not WT, though this did not reach statistical significance. Taken collectively, these data suggest a model whereby Marco-expressing alveolar macrophages may, in responses to an as-of-yet unidentified factor, inhibit *Ace* expression at the gene and protein level, and thereby negatively regulate the cleavage of Angiotensin I to form Angiotensin II, and thereby aldosterone production (*Figure 5E*).

In conclusion, we hereby demonstrate that Marco is a negative regulator of aldosterone production, associated with a suppression of angiotensin-converting enzyme expression in the lungs of male mice. We propose a model in which extra-adrenal Marco expressing alveolar macrophages, through tissue crosstalk with lung endothelial cells, actively inhibit Ace expression and thereby inhibit the production of aldosterone from the adrenal glands.

## Discussion

Multiple studies over preceding decades have shown that macrophages are much more than simply immune and inflammatory cells. Macrophages have been shown to mediate a wide variety of physiological functions and support homeostasis in virtually every tissue they are found (*Theurl et al., 2016*; *Cox et al., 2021*; *Muller et al., 2014*; *Hulsmans et al., 2017*; *Schafer et al., 2012*; *Pirzgalska et al., 2017*). Our original hypothesis centred on the regulation of corticosteroid output via modulation of cholesterol availability. While we ended up disproving this hypothesis, we did provide a novel example of the non-immune functions of tissue macrophages. It was recently shown that tissue-infiltrating macrophages cause neuronal cell death and effective denervation, which disrupts normal melatonin secretion in a model of cardiovascular disease (*Ziegler et al., 2023*). To our knowledge, ours is the first study to report that macrophages are involved in the regulation of hormonal output in vivo at steady state. It is known that immunosuppressants such as cyclosporine can increase blood pressure (*Robert et al., 2010*) yet the mechanism remains incompletely understood. This observation is consistent with our data showing macrophage-mediated negative regulation of the blood pressure-controlling hormone aldosterone. Additionally, the fact that higher ACE/ACE2 ratios have been linked to hypertension and worse outcomes in Covid-19 (*Pagliaro and Penna, 2020*) could implicate MARCO in the susceptibility to severe disease.

Secondary aldosteronism (SA) is distinguished from primary aldosteronism through renin – SA is characterised by high aldosterone and high or non-suppressed renin levels (*Pócsai et al., 2021*). By this definition, Marco-deficient mice exhibit SA, due to their high aldosterone and normal renin levels. SA is also defined as elevated aldosterone that originates outside of the adrenal gland – in this case from ACE in the lung. This finding raises the intriguing possibility that aldosteronism could be triggered or aggravated by MARCO-binding ligands. The fact that blood pressure and plasma electrolytes are not substantially modulated by Marco deletion, despite elevation in plasma aldosterone concentrations, indicates that this mechanism has not evolved to regulate systemic physiology. However, ACE and Angiotensin II are pleiotropic molecules. In addition to their physiological roles, they are also exhibit immunomodulatory functions. In vitro, lipopolysaccharide (LPS) treatment

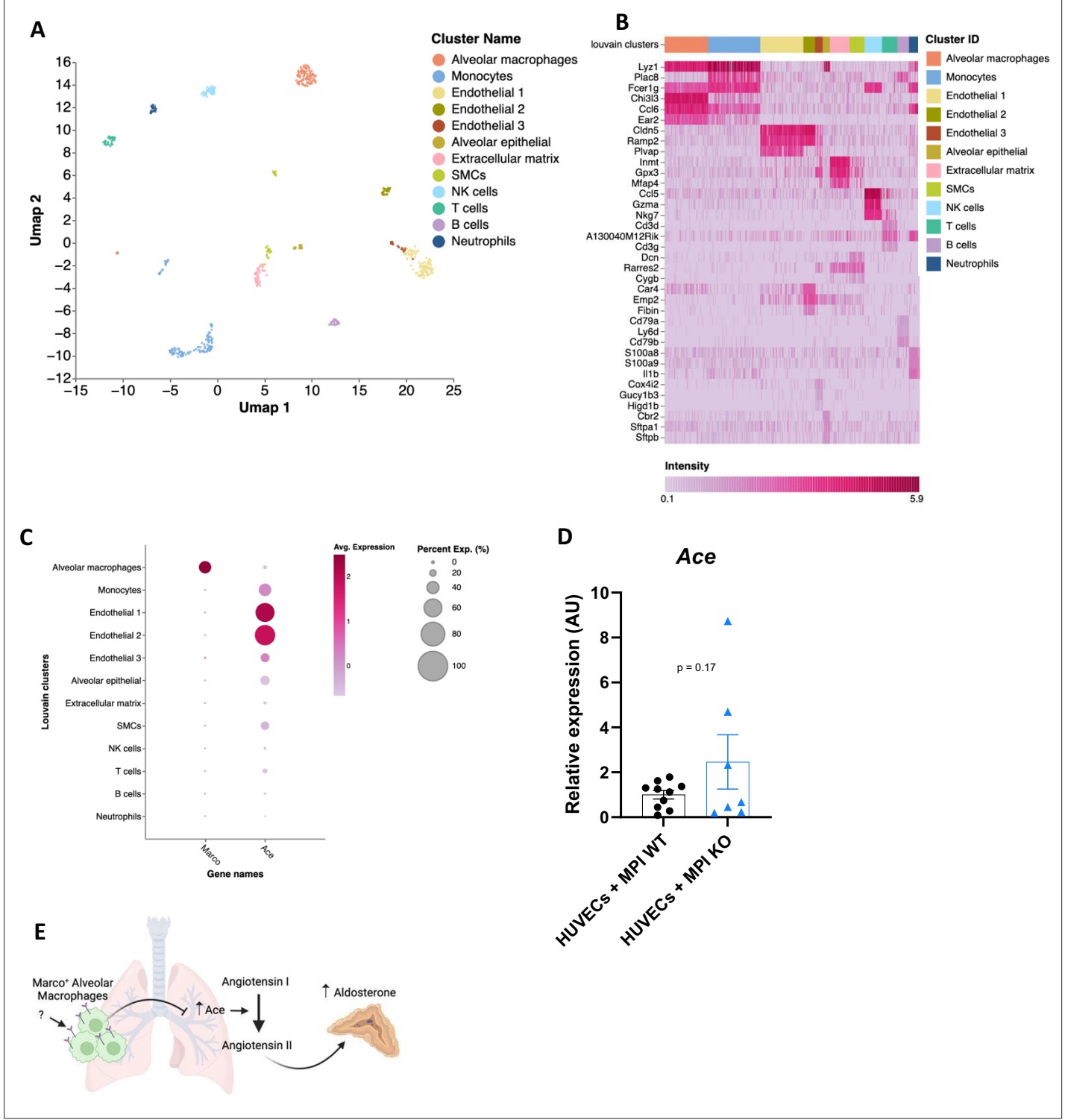

**Figure 5.** A proposed model for macrophage-mediated regulation of lung Ace expression. (**A**) Single-cell RNA-seq UMAP plot depicting the cell types present in the male murine lung. (**B**) Marker heatmap showing the top three gene markers for each cell cluster in (**A**). (**C**) A dot plot showing the gene expression levels of Marco and Ace in the different cell clusters of the male murine lung. (**D**) qPCR data showing the relative expression levels of Ace in co-cultures contraining HUVEC cells with Marco sufficient or deficient MPI macrophages. (**E**) A schematic, generated using BioRender, showing the working model by which Marco+ alveolar macrophages regulate aldosterone output from the adrenal gland. Data in (**D**) were analysed by two-tailed unpaired Student's *t*-test and are shown as average ± SEM.

of the HUVECs endothelial cell line or human primary endothelial cells causes a reduction in ACE activity, mRNA and protein levels. Similarly, ACE expression in human lung tissue is reduced in sepsis patients relative to control tissues (*Hermanns et al., 2014*; *Takei et al., 2019*). Supernatant from LPS or TNFα-treated endothelial cells shows a high proportion of ACE+ endothelial microparticles (EMPs). ACE+ EMPs are also increased in plasma of mice undergoing experimental lung injury and in human septic patients (*Takei et al., 2019*), indicating shedding of lung endothelial ACE in the context of inflammation. Ace and AngII have been shown to enhance inflammation in a variety of contexts. It may therefore be that Marco-expressing alveolar macrophages exert an immune-modulatory function within the murine lung via controlling tissue *Ace* expression. Alveolar macrophages, which use MARCO to bind and uptake bacteria (*Palecanda et al., 1999*), could serve to enhance lung immunity in a pathogen and ACE-dependent manner. Pharmacologic ACE inhibition was previously shown to dampen pro-inflammatory immune cell inflammation in radiation-induced pneumonitis (*Sharma et al., 2022*). Future experiments could therefore explore whether Marco regulates lung tissue Ace expression in the context of inflammation and/or pathogenic challenge. While acute increases of plasma aldosterone have varying effects on endothelial function in humans and animal models, chronic (i.e. genetic) elevations of plasma aldosterone are virtually universally associated with endothelial dysfunction (*Toda et al., 2013*). This is frequently characterised by reduced endothelial NOS activity from, and acetylcholine action on, endothelial cells (*Toda et al., 2013*). It would therefore be logical to test whether vascular dysfunction is present in the endothelia of Marco-deficient mice relative to WT mice.

While the data presented here indicate that Marco deletion is a positive regulator of aldosterone and lung Ace expression, important questions remain to be answered, such as the precise mechanism by which Marco-expressing alveolar macrophages suppress Ace expression, and the ligand(s) responsible for stimulating this effect via the receptor. Additionally, the mechanisms mediating the sex differences observed here remain unexplored. Our findings nevertheless introduce a novel immune paradigm: the renin–angiotensin–aldosterone system.

## Methods

### Mice

C57BL/6J, Marco$^{-/-}$, and hCD68$^{GFP/GFP}$ mice were bred and housed in individually ventilated cages in specific pathogen-free conditions at the University of Oxford. All experiments on mice were conducted according to institutional, national, and European animal regulations. Animal protocols were approved by the animal welfare and ethics review board of the Department of Physiology, Anatomy, and Genetics at the University of Oxford.

### Tissue collection

All tissues were collected from 8- to 11-week-old male mice that were culled between 0930 and 1100. Mice were euthanised by intraperitoneal injection of 10 µl/g 200 mg/ml Pentoject (pentobarbital sodium; Animalcare). Following cessation of the pedal motor reflex, a sample of blood was taken from the right atrium prior to perfusion through the left ventricle with 20 ml 1X phosphate-buffered saline (PBS; Sigma-Aldrich). Adrenal glands were dissected, and the peri-adrenal fat removed using a stereomicroscope. Following harvest, adrenal glands and lungs were frozen at –80°C until subsequent molecular analyses or fixed overnight in 4% paraformaldehyde prior to histological analyses. Bloods obtained prior to perfusion were collected into EDTA coated tubes to prevent clotting and centrifuged at 1000 × *g* for 15 min to separate plasma. Plasma samples were frozen at –80°C in 1.5 ml Eppendorf tubes until subsequent analyses.

### Plasma hormone, cholesterol, renin activity, and electrolyte quantitation

We measured plasma aldosterone and corticosterone using reverse ELISA kits (Enzo Life Sciences) according to the manufacturer's instructions. Plasma cholesterol was quantified by colourimetric assay (Cholesterol Quantitation Kit; Sigma-Aldrich) according to the manufacturer's instructions. Plasma renin activity was measured by fluorometric assay using the Renin Assay Kit (Sigma-Aldrich). Colourimetric and fluorimetric readouts from these assays were attained using a FLUOstar Omega plate reader (BMG Labtech). Plasma electrolytes were outsourced to Pinmoore Animal Laboratory Services

Limited (Cheshire, England) and sodium and potassium electrolyte readings were taken using an I-Smart 30 Vet electrolyte analyser (Woodley Equipment).

## Single-cell sequencing data analysis

Publicly available single-cell RNA-seq datasets (steady-state adrenal: PMID 33571131, GEO accession number GSE161751; steady-state lung: PMID 30283141, GEO accession number GSE109774) were processed, explored and visualised using Cellenics community instance (https://scp.biomage.net/) hosted by Biomage (https://biomage.net/). For the adrenal dataset, the classifier filter was disabled as the sample had been pre-filtered. The cell size distribution filter was disabled. A mitochondrial content filter was used, with the absolute threshold method used (bin step = 0.05, maximum fraction = 0.1). The number of genes vs UMI filter was used with a linear fit and p-value = 0.0004906771. The doublet filter was used (bin step = 0.05, probability threshold = 0. 8743427). The harmony algorithm was used for data integration (2000 genes, log normalisation). The RPCA method was used for dimensionality reduction (13 principal components). Cells and clusters were visualised using UMAP embedding with the cosine distance metric (minimum distance = 0.2). Louvain clustering was the clustering algorithm used in this analysis (resolution = 0.6). For the lung dataset, the classifier filter was disabled as the sample had been pre-filtered. A cell size distribution filter was utilised with binStep = 200 and minimum cell size = 1669. A mitochondrial content filter was used, with the absolute threshold method used, bin step = 0.3, maximum fraction = 0. The number of genes vs UMI filter was used with a linear fit and p-value = 0.001. The doublet filter was used with bin step = 0.02 and probability threshold = 0.7616616. The harmony algorithm was used for data integration (2000 genes, log normalisation). The RPCA method was used for dimensionality reduction (24 principal components). Cells and clusters were visualised using UMAP embedding with the cosine distance metric (minimum distance = 0.3). Louvain clustering was the clustering algorithm used in this analysis (resolution = 0.8).

## Cryo-sectioning and antibody staining

Prior to tissue embedding, adrenal glands were cryoprotected in 30% sucrose solution (in PBS) overnight. Cryoprotected adrenal glands were embedded in OCT embedding medium (Thermo Fisher Scientific) and snap frozen in liquid nitrogen. 15 μm sections were cut from embedded tissues using a Leica cryostat and mounted onto charged microscope slides. Cryosectioned tissue slices were thawed in PBS for 10 min at room temperature, tissue sections were circled using a super PAP pen (Life Technologies), blocked and permeabilised for 1 hr at room temperature in perm/block solution (3% bovine serum albumin, 2% goat serum, 1% Triton X-100, 0.01% $NaN_3$ in PBS). Slides were stained with CD68-AF647 (clone FA-11, BioLegend), chicken anti-GFP (ab13970, Abcam) with goat anti-chicken AF488 (A11039, Invitrogen), mouse anti-MARCO (ED31, Bio-Rad) with goat anti-mouse AF546 (A-11030, Invitrogen), and rabbit anti-ACE (MA5-32741, Invitrogen) with goat anti-rabbit Alexa Fluor 488 Tyramide Super Boost kit (Invitrogen, B40922). Primary stains were done overnight at 4°C and secondary stains were done for 1 hr at room temperature. The goat anti-rabbit Alexa Fluor 488 Tyramide Super Boost kit was used as per manufacturer's instructions. DAPI counterstains were done at 1:1000 for 5 min at room temperature. Fluoromount G mounting medium (Invitrogen) was used to mount coverslips to slides which were then sealed with clear nail varnish. Immunofluorescence images were acquired using the Zeiss 880 confocal microscope and images were analysed in FIJI.

## RT-PCR

Tissues were homogenised using the Precellys Hard Tissue Grinding Kit (MK28-R; Bertin Technologies). Total RNA from homogenised adrenals or lungs was isolated using RNeasy Plus Micro Kit (QIAGEN, cat# 74034). cDNA was reverse transcribed using SuperScript II (Invitrogen) and random primers (Invitrogen). Quantitative PCR was performed using SYBR Green (Applied Biosystems) in C1000 Touch Thermal Cycler (Bio-Rad). Β-actin was used as the housekeeping gene to normalise samples. We used the following formula to calculate the relative expression levels: $RQ = 2^{-\Delta Ct} \times 100 = 2^{-(Ct\ gene\ of\ interest\ -\ Ct\ \beta\ actin)} \times 100$. The following mouse-specific primers were used. *Star* forward 5'-CAGGGCCAAGAAAACCTACA-3'; *Star* reverse 5'-ACGAGCATTTTGAAGCACCT-3'; *Cyp11a1* forward 5'-AGGACTTTCCCTGCGCT-3'; *Cyp11a1* reverse 5'-GCATCTCGGTAATGTTGG-3'; *Hsd3b1* forward 5'-GCGGCTGCTGCACAGGAATAAAG-3'; *Hsd3b1* reverse 5'-TCACCAGGCAGCTCCATCCA-3'; *Cyp11b1* forward 5'-TCACCATGTGCTGAAATCCTTCCA-3'; *Cyp11b1* reverse 5'-GGAAGAGAAGAGAGGGCAATGTGT-3';

*Cyp11b2* forward 5′-CAGGGCCAAGAAAACCTACA-3′; *Cyp11b2* reverse 5′-ACGAGCATTTTGAAGC ACCT-3′; *B-actin* forward 5′-TCATGAAGTGTACGTGGACATCC-3′; *Ace* forward 5′- GCTTCCTCTTTC TGCTGCTCTG-3′; *Ace* reverse 5′-TGCCCTCTATGGTAATGTTGGT-3′; *B-actin* reverse 5′-CCTAGAAG CATTTGCGGTGGACGATG-3′.

## Zona fasciculata suppression model

Water-soluble dexamethasone (Sigma-Aldrich) was reconstituted in autoclaved deionised water at the concentration of 0.0167 mg/ml as described in *Finco et al., 2018*. Ten-week-old *Marco*$^{-/-}$ mice were fed dexamethasone-supplemented drinking water for 14 days prior to blood sampling as described above.

## Digital image analysis

Quantitation of the area and intensities of immunofluorescent stains was done using FIJI image analysis software. Three cryosections of stained lung tissue (~5 $mm^2$ tissue acquired per section) were analysed per mouse. $CD68^+$ cells were segmented using the Otsu thresholding plugin. The parameters were adjusted manually to ensure optimum cell segmentation. DAPI-stained nuclei were segmented using the Otsu thresholding plugin. MFI for ACE was measured via the integrated density of the stain, which was normalised to the area of DAPI staining.

## Blood pressure readings

Systolic, diastolic, and mean blood pressures were taken using the CODA high throughput non-invasive blood pressure system (Kent Scientific). Mice were acclimatised to animal holders on three occasions in the week prior to experimental readings.

## Co-culture

X63-GMCSF and WT and Marco-deficient MPI cells were a kind gift from Dr Subhankar Mukho-padhyay. Cell identity was confirmed by qPCR and lines tested negative for mycoplasma. 20,000 HUVEC cells were plated with 20,000 WT or Marco-deficient MPI cells in 96 well plates for 24 hr in high glucose DMEM, 10% fetal bovine serum, 1% X63-GM-CSF conditioned media as a source of GM-CSF, 1% penicillin/streptomycin (P/S), and 1X MesoEndo Cell growth medium (all bought from Sigma-Aldrich) and incubated at 37°C, 5% $CO_2$. RNA was extracted and qPCRs performed as previously described.

## Statistical analysis

Results are expressed as the mean ± SEM, as indicated in the figure legends. Statistical significance between two experimental groups was assessed using two-tailed Student's *t*-test. All statistical analyses were performed in GraphPad Prism 9 (GraphPad, USA) for Mac OS X. All calculated p-values are reported in the figures, denoted by *p < 0.05, **p < 0.01, ***p < 0.001, ns p > 0.05.

## Acknowledgements

This research was funded in whole or in part by the BHF graduate studentship FS/19/61/34900, the Wellcome/HHMI International Research Scholar award 208576/Z/17/Z, ERC grant 2017 COG 771431, the Pfizer ASPIRE Obesity award and the Next Iteration of the Type 2 Diabetes Knowledge Portal (2UM1DK105554). For the purpose of Open Access, the author has applied a CC BY public copyright licence to any Author Accepted Manuscript (AAM) version arising from this submission

## Additional information

#### Competing interests

Ana I Domingos: Reviewing editor, *eLife*. The other authors declare that no competing interests exist.

## Funding

| Funder | Grant reference number | Author |
|---|---|---|
| British Heart Foundation | FS/19/61/34900 | Conan JO O'Brien |
| Howard Hughes Medical Institute | 208576/Z/17/Z | Ana I Domingos |
| European Research Council | 2017 10 COG 771431 | Ana I Domingos |
| Pfizer | | Ana I Domingos |

The funders had no role in study design, data collection, and interpretation, or the decision to submit the work for publication.

## Author contributions

Conan JO O'Brien, Conceptualization, Data curation, Formal analysis, Supervision, Funding acquisition, Investigation, Visualization, Methodology, Writing – original draft, Project administration, Writing – review and editing; Giorgio Ratti, Emma Haberman, Charles Sweeney, Investigation; Hellen Veida-Silva, Formal analysis, Investigation; Siamon Gordon, Resources, Funding acquisition, Project administration; Ana I Domingos, Funding acquisition, Project administration, Writing – review and editing

## Author ORCIDs

Conan JO O'Brien http://orcid.org/0000-0002-7419-4448
Giorgio Ratti https://orcid.org/0000-0002-9938-2752
Ana I Domingos https://orcid.org/0000-0002-7938-4814

## Ethics

All experiments on mice were conducted according to institutional, national, and European animal regulations. All experimental procedures were performed on living animals in accordance with the UK Animals Acts 1986 under the project licence (PPL no. P80EDA9F7) and personal licences granted by the UK Home Office. Animal protocols were approved by the animal welfare and ethics review board of the Department of Physiology, Anatomy, and Genetics at the University of Oxford.

Reviewer #2 (Public review): https://doi.org/10.7554/eLife.91318.3.sa1
Author response https://doi.org/10.7554/eLife.91318.3.sa2

# Additional files

## Supplementary files

MDAR checklist

Source data 1. This file contains source data for *Figures 1–5*.

## Data availability

All data generated or analysed during this study are included in the manuscript and supporting files.

The following previously published datasets were used:

| Author(s) | Year | Dataset title | Dataset URL | Database and Identifier |
|---|---|---|---|---|
| Lopez JP, Roeh S, Chen A | 2021 | The neuroendocrine stress response at single-cell resolution reveals adrenal ABCB1 as key regulator of stress adaptation | https://www.ncbi.nlm.nih.gov/geo/query/acc.cgi?acc=GSE161751 | NCBI Gene Expression Omnibus, GSE161751 |
| The Tabula Muris Consortium | 2018 | Tabula Muris: Transcriptomic characterization of 20 organs and tissues from *Mus musculus* at single cell resolution | https://www.ncbi.nlm.nih.gov/geo/query/acc.cgi?acc=GSE109774 | NCBI Gene Expression Omnibus, GSE109774 |

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
