## [Editor Report · eLife Assessment]

O'Brien and co-authors provide **important** data demonstrating that tissue-resident macrophages can exert physiological functions and influence endocrine systems. Their model in which AMs negatively regulate aldosterone production via effects exerted in the lung is **solid**. The work will be of broad interest to cell biologists and immunologists.

---

## [Referee Report · Reviewer #2 (Public review)]

Summary:

Tissue-resident macrophages are more and more thought to exert key homeostatic functions and contribute to physiological responses. In the report of O'Brien and Colleagues, the idea that the macrophage-expressed scavenger receptor MARCO could regulate adrenal corticosteroid output at steady-state was explored. The authors found that male MARCO-deficient mice exhibited higher plasma aldosterone levels and higher lung ACE expression as compared to wild-type mice, while the availability of cholesterol and the machinery required to produce aldosterone in the adrenal gland were not affected by MARCO deficiency. The authors take these data to conclude that MARCO in alveolar macrophages can negatively regulate ACE expression and aldosterone production at steady-state and that MARCO-deficient mice suffer from a secondary hyperaldosteronism.

Strengths:

If properly demonstrated and validated, the fact that tissue-resident macrophages can exert physiological functions and influence endocrine systems would be highly significant and could be amenable to novel therapies.

Major weakness:

The comparison between C57BL/6J wild-type mice and knock-out mice for which a precise information about the genetic background and the history of breedings and crossings is lacking can lead to misinterpretations of the results obtained. Hence, MARCO-deficient mice should be compared with true littermate controls.

---

## [Author Response]

The following is the authors’ response to the original reviews

We again thank the reviewers for their comments and recommendations. In response to the reviewer’s suggestions, we have performed several additional experiments, added additional discussion, and updated our conclusions to reflect the additional work. Specifically, we have performed additional analyses in female WT and Marco-deficient animals, demonstrating that the Marco-associated phonotypes observed in male mice (reduced adrenal weight, increased lung Ace mRNA and protein expression, unchanged expression of adrenal corticosteroid biosynthetic enzymes) are not present in female mice. We also report new data on the physiological consequences of increased aldosterone levels observed in male mice, namely plasma sodium and potassium titres, and blood pressure alterations in WT vs Marco-deficient male mice. In an attempt to address the reviewer’s comments relating to our proposed mechanism on the regulation of lung Ace expression, we additionally performed a co-culture experiment using an alveolar macrophage cell line and an endothelial cell line. In light of the additional evidence presented herein, we have updated our conclusions from this study and changed the title of our work to acknowledge that the mechanism underlying the reported phenotype remains incompletely understood. Specific responses to reviewers can be seen below.

**Public Reviews:**

**Reviewer #1 (Public Review):**
Summary:The investigators sought to determine whether Marco regulates the levels of aldosterone by limiting uptake of its parent molecule cholesterol in the adrenal gland. Instead, they identify an unexpected role for Marco on alveolar macrophages in lowering the levels of angiotensin-converting enzyme in the lung. This suggests an unexpected role of alveolar macrophages and lung ACE in the production of aldosterone.Strengths:The investigators suggest an unexpected role for ACE in the lung in the regulation of systemic aldosterone levels.The investigators suggest important sex-related differences in the regulation of aldosterone by alveolar macrophages and ACE in the lung.Studies to exclude a role for Marco in the adrenal gland are strong, suggesting an extra-adrenal source for the excess Marco observed in male Marco knockout mice.Weaknesses:While the investigators have identified important sex differences in the regulation of extrapulmonary ACE in the regulation of aldosterone levels, the mechanisms underlying these differences are not explored.The physiologic impact of the increased aldosterone levels observed in Marco -/- male mice on blood pressure or response to injury is not clear.The intracellular signaling mechanism linking lung macrophage levels with the expression of ACE in the lung is not supported by direct evidence.
**Reviewer #2 (Public Review):**
Summary:Tissue-resident macrophages are more and more thought to exert key homeostatic functions and contribute to physiological responses. In the report of O'Brien and Colleagues, the idea that the macrophage-expressed scavenger receptor MARCO could regulate adrenal corticosteroid output at steady-state was explored. The authors found that male MARCO-deficient mice exhibited higher plasma aldosterone levels and higher lung ACE expression as compared to wild-type mice, while the availability of cholesterol and the machinery required to produce aldosterone in the adrenal gland were not affected by MARCO deficiency. The authors take these data to conclude that MARCO in alveolar macrophages can negatively regulate ACE expression and aldosterone production at steady-state and that MARCO-deficient mice suffer from secondary hyperaldosteronism.Strengths:If properly demonstrated and validated, the fact that tissue-resident macrophages can exert physiological functions and influence endocrine systems would be highly significant and could be amenable to novel therapies.Weaknesses:The data provided by the authors currently do not support the major claim of the authors that alveolar macrophages, via MARCO, are involved in the regulation of a hormonal output in vivo at steady-state. At this point, there are two interesting but descriptive observations in male, but not female, MARCO-deficient animals, and overall, the study lacks key controls and validation experiments, as detailed below.Major weaknesses:(1) According to the reviewer's own experience, the comparison between C57BL/6J wild-type mice and knock-out mice for which precise information about the genetic background and the history of breedings and crossings is lacking, can lead to misinterpretations of the results obtained. Hence, MARCO-deficient mice should be compared with true littermate controls.(2) The use of mice globally deficient for MARCO combined with the fact that alveolar macrophages produce high levels of MARCO is not sufficient to prove that the phenotype observed is linked to alveolar macrophage-expressed MARCO (see below for suggestions of experiments).(3) If the hypothesis of the authors is correct, then additional read-outs could be performed to reinforce their claims: levels of Angiotensin I would be lower in MARCO-deficient mice, levels of Antiotensin II would be higher in MARCO-deficient mice, Arterial blood pressure would be higher in MARCO-deficient mice, natremia would be higher in MARCO-deficient mice, while kaliemia would be lower in MARCO-deficient mice. In addition, co-culture experiments between MARCO-sufficient or deficient alveolar macrophages and lung endothelial cells, combined with the assessment of ACE expression, would allow the authors to evaluate whether the AM-expressed MARCO can directly regulate ACE expression.
**Recommendations for the authors:**

**Reviewer #1 (Recommendations For The Authors):**
(1) Corticosterone levels in male Marco -/- mice are not significantly different, but there is (by eye) substantially more variability in the knockout compared to the wild type. A power analysis should be performed to determine the number of mice needed to detect a similar % difference in corticosterone to the difference observed in aldosterone between male Marco knockout and wild-type mice. If necessary the experiments should be repeated with an adequately powered cohort.

Using a power calculator (https://www.gigacalculator.com/) it was determined that our sample size of 13 was one less than sufficient to detect a similar % difference in corticosterone as was detected in corticosterone. We regret that we unable to perform additional measurements as the author suggested in the available timeframe.

(2) All of the data throughout the MS (particularly data in the lung) should be presented in male and female mice. For example, the induction of ACE in the lungs of Marco-/- female mice should be absent. Similar concerns relate to the dexamethasone suppression studies. Also would be useful if the single cell data could be examined by sex--should be possible even post hoc using Xist etc.

Given the limitations outlined in our previous response to reviewers it was not possible to repeat every experiment from the original manuscript. We were able to measure the expression of lung Ace mRNA, ACE protein, adrenal weights, adrenal expression of steroid biosynthetic enzymes, presence of myeloid cells, and levels of serum electrolytes in female animals. These are presented in figures 1G, 3B, 4A, 4E, 4F, 4I, and 4J. We have elected to not present single cell seq data according to sex as it did not indicate substantial differences between males and females in Marco or Ace expression and so does not substantively change our approach.

(3) IF is notoriously unreliable in the lung, which has high levels of autofluorescence. This is the only method used to show ACE levels are increased in the absence of Marco. Orthogonal methods (e.g. immunoblots of flow-sorted cells, or ideally CITE-seq that includes both male and female mice) should be used.

We used negative controls to guide our settings during acquisition of immunofluorescent images. Additionally, we also used qPCR to show an increase in Ace mRNA expression in the lung in addition to the protein level. This data was presented in the original manuscript and is further bolstered by our additional presentation of expression data for Ace mRNA and protein in female animals in this revised manuscript.

(4) Given the central importance of ACE staining to the conclusions, validation of the antibody should be included in the supplement.

We don’t have ACE-deficient mice so cannot do KO validation of the antibody. We did perform secondary stain controls which confirmed the signal observed is primary antibody-derived. Moreover, we specifically chose an anti-ACE antibody (Invitrogen catalogue # MA5-32741) that has undergone advanced verification with the manufacturer. We additionally tested the antibody in the brain and liver and observed no significant levels of staining.

**Author response image 1. sa2fig1:** 

(5) The link between alveolar macrophage Marco and ACE is poorly explored.

We carried out a co-culture experiments of alveolar macrophages and endothelial cells and measure ACE/Ace expression as a consequence. This is presented in figure 5D and the discussion.

(6) Mechanisms explaining the substantial sex difference in the primary outcome are not explored.

This is outside the scope if this project, though we would consider exploring such experiments in future studies.

(7) Are there physiologic consequences either in homeostasis or under stress to the increased aldosterone (or lung ACE levels) observed in Marco-/- male mice?

We measured blood electrolytes and blood pressure in Marco-deficient and Marco-sufficient mice. The results from these experiments are presented in 4G-4M.

**Reviewer #2 (Recommendations For The Authors):**
Below is a suggestion of important control or validation experiments to be performed in order to support the authors' claims.(1) It is imperative to validate that the phenotype observed in MARCO-deficient mice is indeed caused by the deficiency in MARCO. To this end, littermate mice issued from the crossing between heterozygous MARCO +/- mice should be compared to each other. C57BL/6J mice can first be crossed with MARCO-deficient mice in F0, and F1 heterozygous MARCO +/- mice should be crossed together to produce F2 MARCO +/+, MARCO +/- and MARCO -/- littermate mice that can be used for experiments.

We thank the reviewer for their comments. We recognise the concern of the reviewer but due to limited experimenter availability we are unable to undertake such a breeding programme to address this particular concern.

(2) The use of mice in which AM, but not other cells, lack MARCO expression would demonstrate that the effect is indeed linked to AM. To this end, AM-deficient Csf2rb-deficient mice could be adoptively transferred with MARCO-deficient AM. In addition, the phenotype of MARCO-deficient mice should be restored by the adoptive transfer of wild-type, MARCO-expressing AM. Alternatively, bone marrow chimeras in which only the hematopoietic compartment is deficient in MARCO would be another option, albeit less specific for AM.

We recognise the concern of the reviewer. We carried out a co-culture experiments of alveolar macrophages and endothelial cells and measure ACE/Ace expression as a consequence. This is presented in figure 5D and the implications explored in the discussion.

(3) If the hypothesis of the authors is correct, then additional read-outs could be performed to reinforce their claims: levels of Angiotensin I would be lower in MARCO-deficient mice, levels of Antiotensin II would be higher in MARCO-deficient mice, Arterial blood pressure would be higher in MARCO-deficient mice, natremia would be higher in MARCO-deficient mice, while kaliemia would be lower in MARCO-deficient mice. Similar read-outs could also be performed in the models proposed in point 2.

We measured blood electrolytes and blood pressure in Marco-deficient and Marco-sufficient mice. The results from these experiments are presented in 4G-4M.

(4) Co-culture experiments between MARCO-sufficient or deficient alveolar macrophages and lung endothelial cells, combined with the assessment of ACE expression, would allow the authors to evaluate whether the AM-expressed MARCO can directly regulate ACE expression.

To address this concern we carried out a co-culture experiment as described above.